# Associations between historical redlining and birth outcomes from 2006 through 2015 in California

Anthony L. Nardone[1]*, Joan A. Casey[2], Kara E. Rudolph[3], Deborah Karasek[4], Mahasin Mujahid[5], Rachel Morello-Frosch[5,6]*

1 University of California, Berkeley-University of California San Francisco Joint Medical Program, Berkeley, California, United States of America, 2 Columbia University Mailman School of Public Health, New York, New York, United States of America, 3 Department of Epidemiology, Columbia University, New York, New York, United States of America, 4 Department of Obstetrics, Gynecology, and Reproductive Sciences, University of California, San Francisco, California, United States of America, 5 School of Public Health, University of California, Berkeley, California, United States of America, 6 Department of Environmental Science, Policy and Management, University of California, Berkeley, California, United States of America

* Anthony.nardone@ucsf.edu (ALN); rmf@berkeley.edu (RMF)

**Data Availability Statement:** Per the University of California, Berkeley Institutional Review Board and the Office of Statewide Health Planning and Development's Committee for the Protection of

## Abstract

### Background

Despite being one of the wealthiest nations, disparities in adverse birth outcomes persist across racial and ethnic lines in the United States. We studied the association between historical redlining and preterm birth, low birth weight (LBW), small-for-gestational age (SGA), and perinatal mortality over a ten-year period (2006–2015) in Los Angeles, Oakland, and San Francisco, California.

### Methods

We used birth outcomes data from the California Office of Statewide Health Planning and Development between January 1, 2006 and December 31, 2015. Home Owners' Loan Corporation (HOLC) Security Maps developed in the 1930s assigned neighborhoods one of four grades that pertained to perceived investment risk of borrowers from that neighborhood: green (grade A) were considered "Best", blue (grade B) "Still Desirable", yellow (grade C) "Definitely Declining", and red (grade D, hence the term "redlining") "Hazardous". Geocoded residential addresses at the time of birth were superimposed on HOLC Security Maps to assign each birth a HOLC grade. We adjusted for potential confounders present at the time of Security Map creation by assigning HOLC polygons areal-weighted 1940s Census measures. We then employed propensity score matching methods to estimate the association of historical HOLC grades on current birth outcomes. Because tracts graded A had almost no propensity of receiving grade C or D and because grade B tracts had low propensity of receiving grade D, we examined birth outcomes in the three following comparisons: B vs. A, C vs. B, and D vs. C.

### Results

The prevalence of preterm birth, SGA and mortality tended to be higher in worse HOLC grades, while the prevalence of LBW varied across grades. Overall odds of mortality and

Human Subjects rules, the birth data that support the findings of this study are restricted for transmission to those outside the study investigator team. Data sharing with investigators outside the team requires IRB approval, and requests for birth data may be made directly to the California Department of Public Health's Office of Statewide Health Planning and Development (https://oshpd.ca.gov/data-and-reports/research-data-request-information/ 916-326-3660, cphs-mail@oshpd.ca.gov). The follow data used in this analysis are publicly availably: HOLC map shapefiles from The Mapping Inequality Project (https://dsl.richmond.edu/panorama/redlining/); 1940s census tract data from Individual Public Use Microdata Series (IPUMS) National Historical Geographic Information Systems database (https://www.nhgis.org).

**Funding:** RMF was supported by the National Institutes of Health Environmental influences on Child Health Outcomes (ECHO) program (UG3OD023272 and UH3OD023272) and the UC Berkeley Superfund Research Program (P42ES004705). DK was supported by the Transdisciplinary Postdoctoral Fellowship of the Preterm Birth Initiative at UCSF. JAC was supported by the National Institutes of Environmental Health Sciences R00 ES027023 and P30 ES009089.

**Competing interests:** The authors have declared that no competing interests exist.

preterm birth increased as HOLC grade worsened. Propensity score matching balanced 1940s census measures across contrasting groups. Logistic regression models revealed significantly elevated odds of preterm birth (odds ratio (OR): 1.02, 95% confidence interval (CI): 1.00–1.05), and SGA (OR: 1.03, 95% CI: 1.00–1.05) in the C vs. B comparison and significantly reduced odds of preterm birth (OR: 0.93, 95% CI: 0.91–0.95), LBW (OR: 0.94–95% CI: 0.92–0.97), and SGA (OR: 0.94, 95% CI: 0.92–0.96) in the D vs. C comparison. Results differed by metropolitan area and maternal race.

## Conclusion

Similar to prior studies on redlining, we found that worsening HOLC grade was associated with adverse birth outcomes, although this relationship was less clear after propensity score matching and stratifying by metropolitan area. Higher odds of preterm birth and SGA in grade C versus grade B neighborhoods may be caused by higher-stress environments, racial segregation, and lack of access to resources, while lower odds of preterm birth, SGA, and LBW in grade D versus grade C neighborhoods may due to population shifts in those neighborhoods related to gentrification.

## Introduction

Despite spending more per-capita on healthcare than any other nation in the world, infant, maternal, and neonatal health disparities persist across racial and ethnic groups in the United States (US) [1]. In 2018, pregnant non-Hispanic Black women were one-and-a-half times as likely to deliver a baby preterm and more than twice as likely to deliver a baby of low birth weight (LBW) compared to pregnant non-Hispanic white women, while smaller, yet significant, disparities also exist between Hispanic and non-Hispanic white individuals [2]. Moreover, babies born to pregnant Black women are more than twice as likely as their white counterparts to experience fetal, neonatal, and infant death [3]. One potential driver of racial and ethnic disparities in birth outcomes is residential segregation, through which access to education, income, healthcare, and clean environments is limited and thus, promotes health-diminishing ecosystems [4, 5]. However, limited research has assessed how historical policies of segregation, though now repealed, may perpetuate current health disparities.

Policies geared toward racially segregating neighborhoods in the US were widespread, pervasive, and well-documented in the time following the Great Depression up through the Civil Rights movement [6]. Effects of such policies can outlive their implementation period and may continue to contribute to health disparities even after their repeal [7]. Indeed, studies have shown associations between previous enslavement practices in the southern United States and disparities in heart disease and stroke mortality today [8, 9]. It is imperative to assess the extent to which historically racist policies may continue to be ingrained in government institutions at the federal, state and local levels, and in turn influence current neighborhood structure and shape health disparities.

We examine the role of historical redlining, the practice of categorizing perceived neighborhood mortgage investment risk, on present birth outcomes in three metropolitan areas of California. Historical redlining maps, also called Security Maps were created over 80 years ago for over 200 US cities across the US by the federal Home Owners' Loan Corporation (HOLC), a government body created following the Great Depression in 1934 with the goal of rescuing

homeowners from default by issuing replacement mortgages [10, 11]. These maps shaded neighborhoods one of four colors, corresponding to perceived investment risk, with red indicating the highest risk, hence the term "redlining" [12]. Multiple factors influenced HOLC risk grade, including racial demographics; standardized appraisal forms included input lines of "infiltration of" and "foreign-born" to inform the presence of people of color and immigrants [12].

Though HOLC did not always lend in accordance with Security Map risk grades, HOLC lending did reinforce pre-existing racial segregation; moreover, the legacy of HOLC risk grades are associated with current patterns of neighborhood racial residential segregation, poverty, income inequality, tree canopy coverage, higher ambient temperatures, diesel particulate emissions, and diminished home value appreciation [13–19]. As the practice of redlining segregated and deprived certain communities of investment and opportunities for wealth accumulation through homeownership, it is possible that such policies structured these neighborhoods in ways that affect current health outcomes [17, 19–22]. Ecosocial Theory, developed by Krieger *et al.*, provides a theoretical framework for understanding how redlining and other complimentary racist policies undermine community health. Key to this framework is highlighting "pathways of embodiment" in which an individual's health is a function of individual-level factors, their interaction with institutional power structures, and the historical and community contexts and norms in which these institutions function [7, 23]. These pathways unfold at different spatio-temporal scales and become biologically embodied by altering genetic expression in affected groups and thereby disease distributions across populations. Ecosocial Theory helps elucidate how a history of redlining, which deprived communities of color of property ownership and investment opportunities, in turn drove racial and spatio-temporal patterns of economic deprivation, reduced political power, lack of access to community assets such as greenspace and quality education, and increased exposure to pollution, poorer quality housing, and psychosocial stress. This phenomenon has intergenerational impacts that continue to shape the disparate landscape of power, privilege, and community health today.

Emerging evidence suggests that historical redlining is associated with current patterns of neighborhood-level health disparities. For example, a recent analysis by Krieger *et al.* found an association between historical redlining and risk of preterm birth in New York City, and we previously observed a relationship between redlining and current disparities in rates of asthma-related emergency room visits in California [19, 24]. Both studies conceptualize historical redlining as an indicator of government-sanctioned policies that deprived neighborhoods of capital investment and physically segregated people of color in ways that diminished access to services, amenities, and health care resources that are critical to promoting healthy lifestyles and community well-being [24, 25]. Historical redlining was one of many racist policies in the 1930s and beyond, including racially restrictive covenants, discriminatory insurance underwriting practices by the Federal Housing Association, and predatory lending among others, that segregated neighborhoods [6, 26]. Patterns of these historically discriminatory policies, their lasting legacies, and current social imprint, likely vary regionally across the United States [6, 16, 17].

Accordingly, for this analysis we assessed the relationship between adverse birth outcomes from 2006 to 2015 and HOLC risk grade in Los Angeles (LA), Oakland (OAK), and San Francisco (SF), California. We hypothesized that the odds of adverse birth outcomes would increase as historical HOLC grade worsened. Additionally, we employed propensity score methods using 1940s census tract metrics to restrict the study population to comparable groups among different HOLC-graded neighborhoods and to account for potential unmeasured confounding by factors present when the maps were created. Although Security Maps

were created for eight cities in California, we focused on the three aforementioned metropolitan areas due to availability of 1940s Census data and census tract maps.

## Materials and methods

### Study design

We conducted a retrospective cohort study of all births occurring in California between January 1, 2006 and December 31, 2015. The California Office of Statewide Health Planning and Development supplied the dataset, which included maternal addresses at birth, information on maternal health, race/ethnicity, smoking, educational attainment, and infant gestational age, and sex, as well as fetal growth and neonatal outcomes. By superimposing spatial points of geocoded addresses onto historical HOLC maps, we assigned every birth a HOLC grade of A, B, C or D. Births occurring in neighborhoods shaded in green, considered "Best" by appraisers, received grade 'A'; those shaded in blue, considered "Still Desirable," received grade 'B'; those shaded in yellow, considered "Definitely Declining," received grade 'C'; and those shaded in red, considered "Hazardous" received grade 'D'.

All study design protocols were approved by the Institutional Review Boards of the California Department of Public Health (#13-05-1231) and the University of California, Berkeley (# 2013-10-5693).

### Birth outcome definitions

Our primary perinatal outcomes of interest were preterm birth, LBW, small-for-gestational age (SGA), and perinatal mortality. Our secondary outcomes of interest were very preterm, very low birth weight, and neonatal mortality. Preterm birth and very preterm were defined as a birth after 24 weeks of gestation and before 37 weeks (259 days) and 32 weeks (224 days) of gestation, respectively. We defined LBW and very-LBW as a birth weight less than 2500 grams and less than 1500 grams, respectively, occurring after 24 weeks of gestation. SGA was defined by birth weight less than the US sex-specific 10th percentile of weight for each week of gestation [27]. Mortality was determined using the recorded date of death and was any death that the fetus or baby suffered, excluding elective abortions. Lastly, neonatal mortality was defined as any death of a baby born after 24 weeks of gestation within the first 28 days of life. Births with unknown outcomes were excluded from our analyses. All births before occurring before 24 weeks of gestation were dropped from the analysis.

### HOLC maps and grade assignment

Shapefiles of HOLC maps were downloaded from the University of Richmond Virginia's Mapping Inequality Project [28]. Maps from Los Angeles (LA), Oakland (OAK), and San Francisco (SF), California were merged into one shapefile and overlaid onto geocoded addresses to assign a HOLC grade to each birth. Births that that fell outside of categorized regions but within the boundaries of a HOLC map were assigned a grade of 'NG' for "not graded". All births outside of the map boundaries were fully excluded from the analysis. See Fig 1 for more information on exclusion criteria.

### Areal apportionment of 1940s census tract data

Following Aaronson *et al* and Jacoby *et al*, we adjusted this analysis for 1940s census tract measures, as current measures likely mediate the relationship between 1940s measures and current health [16, 29]. HOLC grades led to reduced home ownership and wealth-generating opportunities in redlined and "yellow-lined" communities, which we hypothesize could subsequently

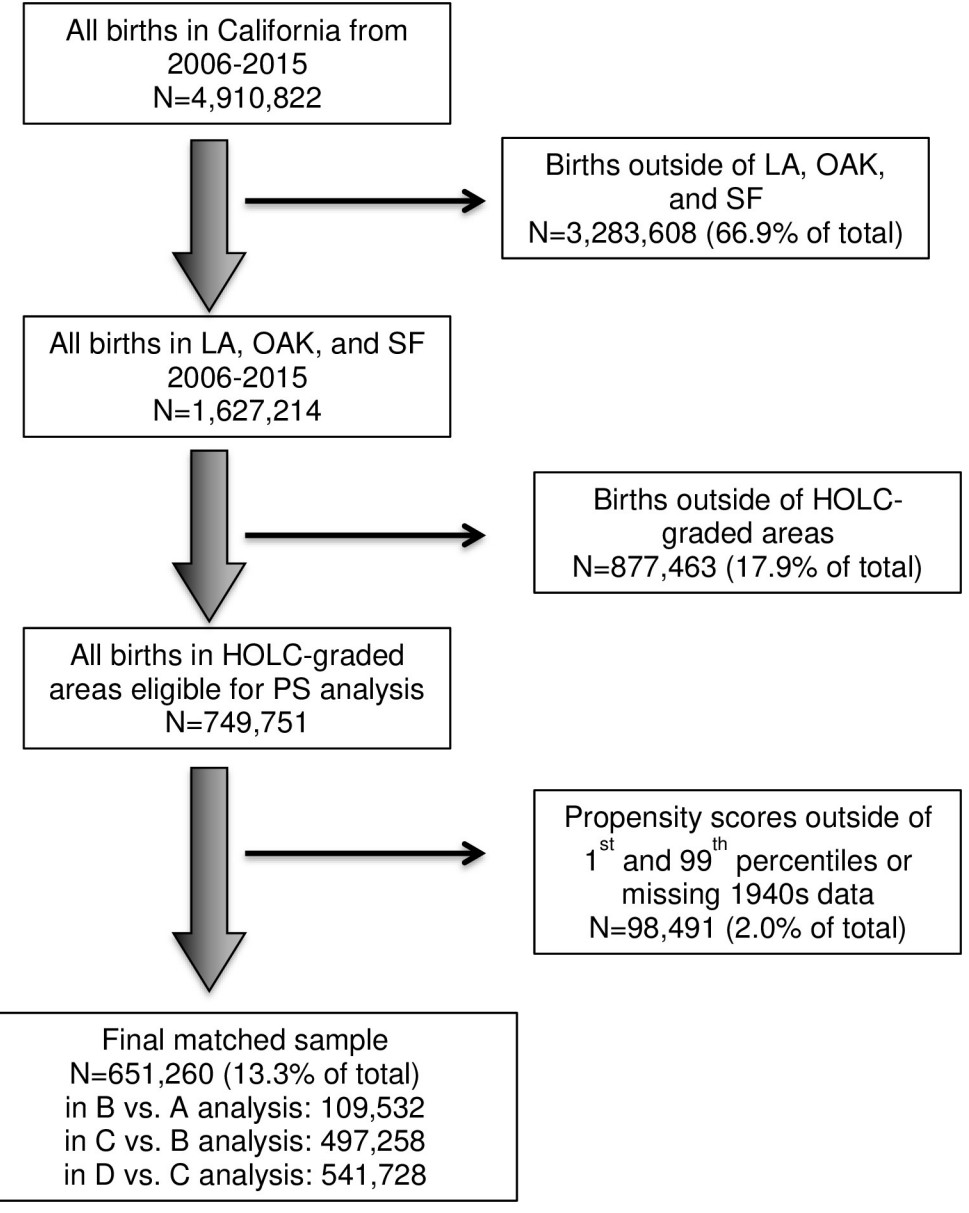

**Fig 1. Inclusion/exclusion criteria flow chart.**

lead to worse current birth outcomes. As such, present day variables would act as mediators and therefore bias our results [30]. As the first HOLC maps were created beginning in 1935, we chose to use 1940s-census measures which were publicly available from the Individual Public Use Microdata Series (IPUMS) National Historical Geographic Information Systems database [31]. Variables from the 1940s census that we included in our analyses were total number of White, non-White, foreign-born White, Black, and employed residents, the number of people per housing unit, number of total homes and homes needing major repairs, the number of homes with and without refrigerators, radios, and heating, and the number of residents at different education levels. We additionally calculated population density by dividing the number of residents within the 1940s census tract by the area of the 1940s census tract.

HOLC maps of LA, OAK, and SF were intersected with 1940s census tract maps using the *rgeos* package in R, with the goal of assigning 1940s census metrics to every HOLC-designated neighborhood [32]. As the boundaries of the HOLC maps and 1940s census tract maps did not overlap perfectly, we assigned 1940s metrics to the HOLC neighborhoods using areal apportionment. The area of each 1940s census tract that overlapped any HOLC polygon was calculated, and subsequently, the proportion of the census tract area of overlap to the total HOLC neighborhood area was calculated. This proportion was then used to weight the 1940s census metrics. For example, if a HOLC neighborhood was intersected by two 1940s census tracts, one comprising 20% of the total HOLC neighborhood area and the other 80%, the corresponding 1940s census tract metrics were multiplied by the overlap proportion (0.2 and 0.8, respectively), and then added together to create the areal-weighted census measure for that HOLC neighborhood.

### Propensity score matching

Following assignment of weighted 1940s census measures to HOLC polygons, we estimated propensity scores using SuperLearner with mean models, generalized linear models, Bayesian generalized linear models, multivariate adaptive regression splines, and generalized additive model algorithms [33]. All 1940s census variables enumerated above were used as covariates. We calculated three sets of propensity scores for the following comparisons: (1) propensity of a neighborhood being grade B among grade A and B neighborhoods; (2) propensity of being grade C among grade C and B neighborhoods; (3) propensity of being grade D among grade D and C neighborhoods. We pursued this approach to avoid practical violations of the positivity assumption (Fig 2A).

Once calculated, neighborhoods with propensity scores below the 1st percentile of the better-graded group or above the 99th percentile of the worse-graded group were excluded from the analysis to avoid relying on extrapolation (Fig 1) [34]. This restricted sample formed the basis of our primary analysis. Following exclusion of off-support or near-off-support geographies, we conducted full matching propensity score on the remaining full sample to construct each pair of HOLC comparison groups. We used the R Package 'MatchIt' to match the worse-graded (treatment) neighborhoods to better-graded (control) neighborhoods based on full matching with replacement [35]. Full matching is a flexible matching algorithm similar to subclassification with many subclasses in which each subclass has either 1) one "exposed" geography and multiple "unexposed" geographies or 2) one "unexposed" geography and multiple "exposed" geographies [36]. "Exposed" and "unexposed" HOLC grades are weighted depending on the number in each subclass. Control matches were required to be within 0.2 standard deviations of the treatment neighborhood, with Mahalanobis-metric matching on the percent of non-White residents and median home value. Control neighborhoods could be matched to more than one treatment neighborhood, and treatment neighborhoods could be matched to more than one control neighborhood. This produced HOLC-neighborhood propensity matching weights for the B vs. A, C vs. B, and D vs. C comparisons, which were then assigned to each birth in the dataset based on residential address. These weights were applied in the analyses that follow.

### Statistical analysis

All statistical analyses were conducted in R. Weighted post-matching comparisons of neighborhood 1940s census metrics were made via weighted *t*-tests with the R package *weights* [37]. Logistic regression models were weighted using propensity score weights produced with MatchIt and adjusted for the following areal-weighted 1940s metrics: median home value,

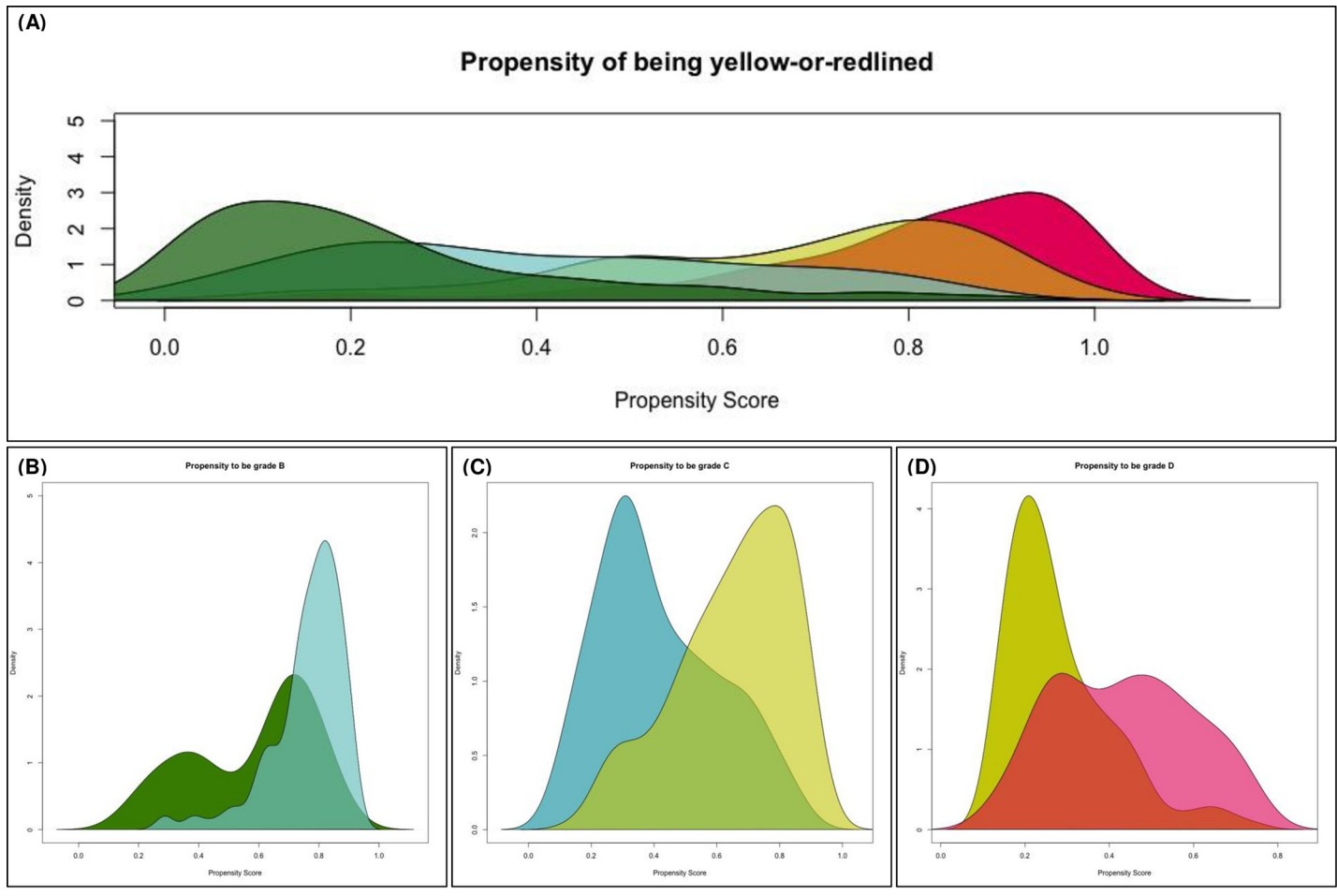

**Fig 2. Propensity score histograms.**

percent of population employed, percent of radio ownership, and percent of non-white residents. Covariates that were not highly collinear based on Spearman's rho (S1 Table) were selected for model inclusion. As we produced three sets of matched propensity score weights in order to compare grades B to A, C to B, and D to C, each odds ratio represents a different logistic regression model including only the births occurring in the matched neighborhoods. We subsequently conducted sensitivity analyses adjusting these models for maternal age, birth year, and maternal parity (S2 Table).

## Results

In total, 1,627,214 births occurring between 2006–2015 in the LA, OAK, and SF metropolitan areas were screened for inclusion eligibility, 651,260 (40.0%) of which fell inside HOLC-graded regions and were included in the final analysis. The proportion of births delivered by women self-identifying as non-Hispanic (NH) Black and Hispanic was 5.1% and 12.1%, respectively, in grade A neighborhoods and 8.9% and 67.2%, respectively, in grade D neighborhoods, while the proportion of births delivered by women identifying as NH-Asian-Pacific Islander (NH-API), NH-White, and NH-Other all decreased as HOLC grade worsened (Table 1). Mean maternal age decreased as HOLC grade worsened from 33.8 (±5.2 years) to 29.1 (±6.3 years). Pregnant women in previously redlined neighborhoods were more likely to pay for

**Table 1. Socio-demographic and delivery characteristics of births by HOLC grade In San Francisco, Oakland and Los Angeles California.**

|  |  | N = 162,7214 A (N = 19,871) | B (N = 99,641) | C (N = 399,770) | D (N = 230,469) | NG (N = 877,463) | p-value |
|---|---|---|---|---|---|---|---|
| Race of mother | Hispanic | 2403 (12.1%) | 35777 (35.9%) | 231036 (57.8%) | 154770 (67.2%) | 497396 (56.7%) | <0.0001 |
|  | NH API | 4190 (21.1%) | 18012 (18.1%) | 55320 (13.8%) | 22263 (9.7%) | 158835 (18.1%) |  |
|  | NH Black | 1017 (5.1%) | 7329 (7.4%) | 34892 (8.7%) | 20450 (8.9%) | 40042 (4.6%) |  |
|  | NH Other | 980 (4.9%) | 4051 (4.1%) | 10724 (2.7%) | 6017 (2.6%) | 20177 (2.3%) |  |
|  | NH White | 11281 (56.8%) | 34472 (34.6%) | 67798 (17.0%) | 26969 (11.7%) | 161013 (18.3%) |  |
| Mean age of mother (years) |  | 33.8 (5.2) | 31.2 (6.2) | 28.8 (6.4) | 28.2 (6.5) | 29.1 (6.3) | <0.0001 |
| Hospital/clinic births |  | 19606 (98.7%) | 98494 (98.8%) | 397464 (99.4%) | 229084 (99.4%) | 874457 (99.7%) | <0.0001 |
| Payment type | Private | 16874 (84.9%) | 62326 (62.6%) | 153563 (38.4%) | 70980 (30.8%) | 403558 (46.0%) | <0.0001 |
|  | Medi-Cal | 1776 (8.9%) | 32553 (32.7%) | 228566 (57.2%) | 150564 (65.3%) | 414513 (47.2%) |  |
|  | Other | 363 (1.8%) | 930 (0.9%) | 5455 (1.4%) | 4623 (2.0%) | 20928 (2.4%) |  |
|  | OOP | 806 (4.1%) | 33598 (3.6%) | 11216 (2.8%) | 3690 (1.6%) | 36411 (4.1%) |  |
|  | Unknown | 52 (0.3%) | 234 (0.2%) | 970 (0.2%) | 611 (0.3%) | 2051 (0.2%) |  |
| Adequate care | Yes | 9028 (45.4%) | 45292 (45.5%) | 180806 (45.2%) | 104080 (45.2%) | 420509 (47.9%) | <0.0001 |
| Maternal Education | No HS D | 402 (2.0) | 13902 (14.0) | 118432 (29.6) | 84115 (36.5) | 208252 (23.7) | <0.001 |
|  | HS only | 3328 (16.7) | 29982 (30.1) | 154378 (38.6) | 91104 (39.5) | 354430 (40.4) |  |
|  | Associate's | 847 (4.3) | 5119 (5.1) | 17214 (4.3) | 8296 (3.6) | 48692 (5.5) |  |
|  | Bachelor's | 7613 (38.3) | 27274 (27.4) | 61010 (15.3) | 25592 (11.1) | 160773 (18.3) |  |
|  | Masters+ | 7037 (35.4) | 20458 (20.5) | 37801 (9.5) | 15630 (6.8) | 86489 (9.9) |  |
|  | Unknown | 644 (3.2) | 2906 (2.9) | 10935 (2.7) | 5732 (2.5) | 18827 (2.1) |  |
| WIC Use | Yes | 1883 (9.5) | 33439 (33.6) | 225625 (56.4) | 146227 (63.4) | 429656 (49.0) | <0.0001 |
| Metropolitan area | Los Angeles | 15331 (77.2) | 74340 (74.6) | 347234 (86.9) | 186015 (80.7%) | 832137 (94.8%) |  |
|  | Oakland | 3022 (15.2) | 11409 (11.5) | 29639 (7.4) | 20127 (8.7%) | 20687 (2.4%) | <0.0001 |
|  | SF | 1518 (7.6) | 13892 (13.9) | 22897 (5.7) | 24327 (10.6%) | 24639 (2.8%) |  |

Abbreviations: API-Asian-Pacific Islander; NG-not graded; NH-non-Hispanic; Private-private insurance; SF-San Francisco; HS D-high school degree; HS-high school; SF-San Francisco; hosp/clinic-hospital or clinic; OOP-out of pocket; Masters+-Masters degree or higher.

All births were included if parental residential address was within the boundaries of a HOLC security map. P-values were calculated via X$^2$-tests for categorical variables and ANOVA for continuous variables.

delivery care with Medi-Cal (65.3%), receive the Special Supplemental Nutrition Program for Women, Infants, and Children (WIC) (63.4%), and were less likely to have a bachelor's degree (11.1%) compared to those living in grade A neighborhoods (8.9%, 9.5%, and 38.3%, respectively). Overall, the prevalence of preterm birth, SGA, and mortality were significantly higher in Grade C and D neighborhoods compared to grade A neighborhoods, while the prevalence of LBW varied more across HOLC grades (Fig 3).

Distributions of the propensity of being grade C or grade D overall are shown in Fig 2A, along with distributions of propensity scores for the individual comparisons (B vs. A, C vs. B, and D vs. C). Table 2 shows comparisons of 1940s census measures between the groups pre- and post-propensity score matching. Prior to propensity score restriction and matching, median home value and percent non-White residents significantly differed across all comparisons (B vs. A, C vs. B, and D vs. C), along with other census measures. After propensity score restriction and matching, no significant differences remained across any 1940s census measures in these three groups.

Odds ratios calculated from adjusted logistic regression using propensity score matching weights (Table 3) revealed slightly increased odds of preterm birth and SGA in the C vs. B comparison and slightly decreased odds of preterm birth, LBW, and SGA in the D vs. C comparison. Stratifying this analysis by metropolitan area (LA and OAK/SF) revealed area-specific

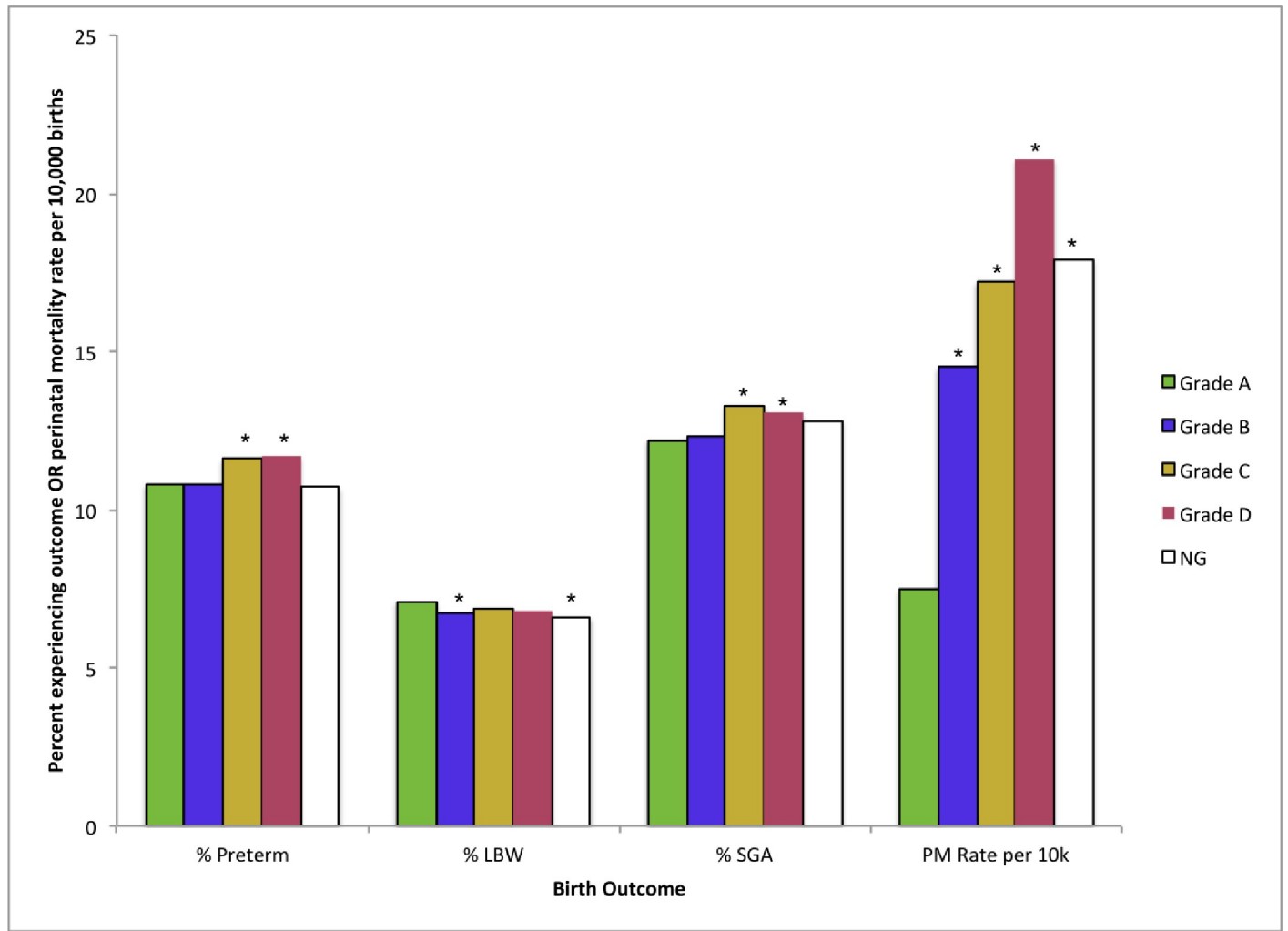

**Fig 3. Birth outcomes by HOLC grade.**

associations between HOLC grade and birth outcomes. Similar to the overall analysis, historical redlining (grade D) in Los Angeles was associated with lower odds of preterm birth, SGA, and LBW compared to births in grade C neighborhoods (Table 3). For births in the OAK/SF

**Table 2. Mean differences of 1940s areal-weighted census metrics pre-and-post propensity score matching.**

|  | Pre-PS Match | | | Post-PS Match | | |
|---|---|---|---|---|---|---|
|  | **B vs. A** | **C vs. B** | **D vs. C** | **B vs. A** | **C vs. B** | **D vs. C** |
| MHV | -2.3 (-3.0, -1.5) | -1.3 (-1.6, -0.9) | -1.1 (-1.4, 0.8) | 0.0 (-0.6, 0.5) | 0.2 (-0.1, 0.5) | 0.0 (-0.3, 0.3) |
| % Major Repairs | 1.0 (-0.1, 2.0) | 0.8 (0.0, 1.5) | 3.4 (1.8, 5.0) | -0.2 (-1.0, 0.7) | 0.6 (-0.2, 1.4) | 1.1 (-0.4, 2.5) |
| % HS Educated | -7.5 (-15.6, 0.6) | -0.5 (-7.0, 5.9) | 2.2 (-3.7, 8.1) | 0.6 (-4.8, 6.1) | 0.4 (-3.7, 4.5) | 2.8 (-5.1, 10.6) |
| % Non-white | -1.0 (-1.6, -0.5) | 0.4 (0.0, 0.7) | 3.4 (1.7, 5.0) | 0.0 (-0.4, 0.4) | 0.0 (-0.4, 0.4) | -0.4 (-2.2, 1.5) |
| % Employed | -22.4 (-36.4, -8.4) | -2.5 (-7.8, 2.8) | -0.4 (-6.4, 5.6) | 0.7 (-4.9, 6.4) | 0.3 (-3.9, 4.6) | 1.5 (-4.4, 7.3) |
| % Radio Ownership | 3.5 (0.4, 6.6) | -0.2 (-1.9, 1.4) | -2.8 (-4.8, -0.8) | -0.2 (-4.0, 3.5) | -1.6 (-3.9, 0.6) | 0.4 (-4.3, 5.2) |
| % Refrigerator Ownership | 12.9 (-0.4, 26.1) | -1.7 (-13.1, 9.6) | 6.5 (-14.4, 27.3) | -0.7 (-2.3, 0.8) | 1.3 (0.4, 2.2) | 0.3 (-0.4, 1.1) |

Abbreviations: MHV-median home value; HS-high school; PS-Propensity Score.

**Table 3. Odds ratios of birth outcomes across multiple comparisons and cities following propensity score matching.**

| | | B vs. A | C vs. B | D vs. C |
|---|---|---|---|---|
| | | OR (95% CI) | OR (95% CI) | OR (95% CI) |
| **All births** | Preterm birth | 1.02 (0.94, 1.10) | 1.02 (1.00, 1.05) | 0.93 (0.91, 0.95) |
| | LBW | 0.90 (0.82, 0.98) | 1.02 (0.99, 1.06) | 0.94 (0.92, 0.97) |
| | SGA | 0.94 (0.88, 1.01) | 1.03 (1.00, 1.05) | 0.94 (0.92, 0.96) |
| | Mortality | 0.69 (0.41, 1.16) | 1.13 (0.92, 1.40) | 1.08 (0.91, 1.28) |
| **LA only** | Preterm birth | 1.02 (0.94, 1.11) | 1.03 (1.00, 1.06) | 0.94 (0.91, 0.97) |
| | LBW | 0.87 (0.80, 0.96) | 1.01 (0.98, 1.04) | 0.95 (0.92, 0.98) |
| | SGA | 0.94 (0.88, 1.01) | 0.98 (0.96, 1.01) | 0.95 (0.93, 0.98) |
| | Mortality | 0.64 (0.37, 1.12) | 1.04 (0.82, 1.31) | 1.07 (0.86, 1.32) |
| **SF-OAK only** | Preterm birth | 1.00 (0.74, 1.34) | 1.10 (1.01, 1.19) | 1.02 (0.96, 1.09) |
| | LBW | 0.97 (0.69, 1.36) | 1.09 (1.00, 1.20) | 0.96 (0.89, 1.03) |
| | SGA | 0.90 (0.70, 1.15) | 1.04 (0.97, 1.12) | 1.04 (0.98, 1.09) |
| | Mortality | 1.51 (0.10, 23.8) | 1.49 (0.82, 2.72) | 1.19 (0.77, 1.85) |

Abbreviations: OR-odds ratio; CI-confidence interval; LA-Los Angeles metropolitan area; Mortality-perinatal mortality; SF-OAK-San Francisco-Oakland metropolitan area.

All models were adjusted for 1940s median home values, percent employed residents, percent non-white residents, and percent with radio ownerships.

metropolitan area, grade C was associated with increased odds of preterm birth and LBW (Table 3) compared to grade B; no significant associations were found between grade D and C in OAK/SF. Sensitivity analysis adjusting for maternal covariates of maternal age, parity, and year of birth did not significantly alter results (S2 Table). Analyses also revealed associations in the grade C versus B comparison for very preterm birth, very LBW, and NND (S3 Table) and reduced odds for very preterm in the grade D versus C comparison. Lastly, analyses of birth outcomes by maternal race and HOLC grade is shown in S4 Table, and unrestricted analysis in S5 Table.

## Discussion

At a time when the American economy was recovering in the aftermath of the Great Depression, HOLC, along with other federal, state, and local government entities, embraced policies that reinforced and perpetuated racial residential segregation. The effects of these discriminatory policies may linger and influence the socioeconomic and structural landscapes of neighborhoods today [38]. We sought to assess if investment grades assigned by HOLC Security maps, tools created for cities across the country that designated investment risk, in-part, as a function of neighborhood racial demographics in the 1930s and 1940s, are associated with current adverse birth outcomes across a ten year period (2006–2015) in San Francisco, Oakland and Los Angeles, California. This paper makes three key contributions to the literature: (1) We found overall elevated odds of adverse birth outcomes in neighborhoods graded as higher investment risks by HOLC Security Maps. (2) By employing propensity score matching of HOLC-graded neighborhoods based on 1940 Census metrics, we found evidence that being assigned a grade C, deemed "definitely declining" by appraisers, compared to B, was associated with increased odds of preterm birth, SGA, and LBW suggesting a potential adverse effect of "yellow-lining" [16]. (3) We also found slightly lower odds of preterm birth, LBW, and SGA when comparing redlined neighborhoods (Grade D) to "yellow-lined"(Grade C) neighborhoods.

To our knowledge, this is the first study to examine relationships between historical redlining and current patterns of adverse birth outcomes in California. In a recent study, Krieger *et al*. found increased odds of preterm birth among 528,096 births in New York City between 2013 and 2017 in historically redlined areas compared to the best-graded areas, though this association was statistically insignificant after adjusting for present-day racialized economic segregation [24]. Though we similarly found higher odds of preterm birth in grade 'C' neighborhoods, several key differences between our analyses hamper study comparisons. First, we used propensity score methods to compare adjacent HOLC grades as opposed to using births in grade 'A' neighborhoods as the reference group, as done by Krieger et al. Second, we adjusted for 1930s Census demographics, while Krieger et al. adjusted models for contemporary census tract-level and individual-level measures. We chose not to adjust for present-day factors as they likely represent mediators in the model.

Other studies have examined the birth outcome effects of modern forms of redlining. For example, in a study in Philadelphia, Mendez *et al* defined redlined neighborhoods as those where Black applicants were more likely to be denied mortgages compared to White applicants between 1999 and 2004, and found similarly reduced odds of preterm birth in redlined areas, specifically among Black women [39]. We found similarly reduced odds of preterm birth along with SGA and LBW when comparing births in historically redlined neighborhoods compared to yellow-lined neighborhoods, an effect that was most predominant among Hispanic women and non-Hispanic White women. The association of historical redlining with several other health outcomes have been shown in other studies; our prior work found a 39% increase in asthma emergency department visit rates in redlined California census tracts compared to grade A census tracts [19]. Other studies have identified relationships between historical redlining and current foreclosures on one hand, and self-rated health on the other in Detroit, Michigan as well as tuberculosis cases in Austin, Texas in the 1950s [20, 21].

Although the directionality of the relationships we identified between historical redlining and birth outcomes overall coincide with prior studies, application of propensity score methods on 1940s Census measures yielded mixed findings. Furthermore, our approach differs from prior HOLC-health studies by only comparing health outcomes in neighborhoods that were most similar when Security Maps were created. Restricting our propensity score analysis by eliminating births occurring in HOLC neighborhoods with propensity scores below the 1st percentile of the better-graded neighborhoods and above 99th percentile of the worse-graded neighborhoods reduced our sample size and thus restricted the geographic generalizability of our results. Analyses of unrestricted data (no percentile-based exclusion), though subject to bias, resulted in findings of similar directionality with an increased strength of association. Though the propensity score-restricted findings reduce generalizability, they reduce bias by not making comparisons beyond what the data can support.

Aaronson *et al* also applied propensity score methods with D vs. C and C vs. B comparisons in their study of the effect of HOLC grades over time on home value, credit score, home ownership, and racial segregation trajectories, although with many differences [16]. Nonetheless, our study builds on their prior work and suggests that, given the pre-existing differences in neighborhoods when Security Maps were created, future analyses should consider these confounders if attempting to make causal inference. In other words, direct comparison of grade D to grade A and B places, though important in identifying current health and resource disparities across cities, may not accurately reveal the extent to which historical redlining shaped such disparities. Multiple studies have also identified associations between racial residential segregation and poor birth outcomes across the United States [40, 41]. A 2016 meta-analysis that assessed the relationship between five measures of segregation and birth outcomes found that black mothers living in communities designated as segregated by at least four of five metrics

had significantly elevated risks of preterm birth and LBW, but that effect estimates differed across segregation metrics [42].

In conjunction with segregation, other neighborhood-level socioeconomic factors, environmental factors, and novel indicators of structural racism that resulted from redlining patterns are also associated with preterm birth and LBW disparities [43–46]. Redlining and "yellow-lining" reduced subsequent home ownership rates and home value appreciation, factors associated with current poverty, and thus likely contribute to the current racial wealth gap in the United States [15, 16, 47]. Limited access to retail services like supermarkets in previously redlined and yellow-lined neighborhoods may also contribute to the relationships we identified [48–50]. Elevated and chronic exposure to psychosocial stressors in previously these neighborhoods may also predispose individuals to poor birth outcomes [4, 51]. Jacoby *et al* identified a stepwise-association between recent firearm assaults and HOLC grade in Philadelphia, suggesting that places assigned worse HOLC grades have higher present day crime rates [29]. Coupling crime to birth outcomes, Matoba *et al* found that preterm birth, LBW, and SGA occurred more frequently over a 5-year period in Chicago neighborhoods where firearm assault was more frequent and found that these outcomes were more frequent among non-Hispanic black women regardless of violence tertile, which the authors attributed to other segregation-associated stressors [52].

We found significantly lower odds of preterm birth, LBW, and SGA in the grade D versus grade C comparison, findings that were robust to additional sensitivity analyses. This is possibly a by-product of increased social cohesion and resilience in previously redlined places compared to yellow-lined places. The evidence linking birth outcomes to living in ethnic enclaves is mixed, though health benefits of living in such places are typically attributed to enhanced social ties and support [53–55]. Gentrification, a process occurring in many previously redlined places, may affect our results by disrupting social networks and fostering higher-stress environments, though evidence for this is mixed [56–61]. Evidence of the benefits of social cohesion and resilience on birth outcomes is mixed, though one study in Los Angeles found that low resilience, measured by validated self-esteem and empowerment questionnaires, was associated with a 12% increased risk of preterm birth [62–65]. On the other hand, though we found that "yellow-lining" was associated with higher odds of preterm birth and SGA compared to grade B neighborhoods, the lack of a similar relationship between grades D and C neighborhoods could potentially be explained by different spatial and demographic distribution of segregation between these grades [66].

Our metropolitan area stratification also revealed unique area-specific results; although odds of preterm birth, LBW, and SGA were similarly reduced in LA, findings from OAK-SF do not follow the same pattern. This further demonstrates the need to assess the legacy of HOLC Security Maps regionally, as differences in federal appraisal practices and local practices may explain this divergence [67]. This result may be partially due to differences in demographics across the two metropolitan areas. The majority of births were to Hispanic mothers in previously redlined (73.1%) and yellow-lined (63.3%) neighborhoods of LA, significantly higher than in the OAK-SF area (37.4% and 22.8% respectively).

A limitation of our study is the possibility of selection bias introduced by the limited spatial coverage of HOLC maps; births occurring outside of graded areas but within the confines of the redlining maps were excluded from this study. As these neighborhoods were not graded by HOLC appraisers, the structural and socioeconomic development of these places are not likely comparable to those that received a HOLC grade [16]. As such, our findings are only generalizable to people living within neighborhoods that received a HOLC grade and did not have outlier propensity scores in California. From 2006 through 2015, births that occurred in places receiving a HOLC grade accounted for just 17.8% of total births. Although we had birth

outcome data from across California, including redlined cities such as Fresno, Sacramento, San Diego, San Jose, and Stockton, 1940s census tract level data for these other areas was not publicly available, and therefore, births from these places were excluded, limiting generalizability.

Another limitation of our analysis is the lack of dataset identifiers to account for births delivered by the same parent over time. We used address at the time of birth to assign births to neighborhoods, however the length of time that each individual lived at that address is unknown. Length of residence in historically redlined or yellow-lined places may be a more relevant exposure metric for birth outcomes. Considering residence only at the time of birth discounts the potential effect of residence in a redlined neighborhood of the life course, including, for example, effects on maternal lifetime stress, which could influence birth outcomes [68]. On the other hand, mobility during pregnancy is itself a risk factor for adverse birth outcomes, and may have been impacted by processes displacement [69]. Additionally, we evaluated multiple birth outcomes and HOLC grade comparisons, which potentially increased the probability of Type I error. Finally, pregnant individuals comprising the categories "Black" "Hispanic" and "Asian" are heterogeneous, comprised of many subgroups (e.g., Mexican, Puerto Rican), and therefore, we may be missing important sub-group specific effects.

Our analysis differs from prior HOLC-related health studies in terms of population, sample size, and methodology: our study population a longer study period (10 years) and a larger sample size compared to previous redlining studies. We applied propensity score matching to enhance the robustness of our results by only comparing health outcomes in neighborhoods that were most similar when Security Maps were created.

## Conclusion

Achieving health equity across racial/ethnic and class lines require better understanding of how historically discriminatory policies, such as redlining, continue to shape current patterns of health disparities across the United States. Our findings indicate that births in "yellow-lined" neighborhoods deemed "Definitely Declining" places for investment in the 1930s and 40s have increased odds of preterm birth, LBW, and SGA, compared to grade "B" neighborhoods, while redlined neighborhoods have lower odds of these outcomes compared to grade "C" neighborhoods. Future studies on the current health effects of historical redlining should expand the geographic scope of analysis and apply rigorous approaches to address unmeasured confounding, such propensity scoring methods, to address challenges with model extrapolation and bias.

## Supporting information

**S1 Table. Spearman's correlation coefficient matrix of 1940s areal apportioned covariates across all births.** Abbreviations: MHV-median home value; %HS Educated-percentage of residents who completed high school, some college, or all of college; %-percent. All correlation coefficients were statistically significant (p<0.05).
(DOCX)

**S2 Table. Propensity score matched analysis odds ratio sensitivity analysis.** Model 1 was adjusted for MHV, % employed, %non-white, and % with radio ownership; Model 2 was additionally adjusted for maternal age, parity, and birth year.
(DOCX)

**S3 Table. Propensity score matched analysis including very preterm (<32 weeks), very low birth weight (<1500 grams), and neonatal death.** Abbreviations: VPT-very preterm; VLBW-

very low birth weight; NND-neonatal death; OR-odds ratio; CI-confidence interval. Odds ratios were derived from propensity score matched analysis.
(DOCX)

**S4 Table. Odds of birth outcomes by HOLC grade comparison and maternal race.** Models were adjusted for 1940s median home value, percent of employed population, percent non-white and foreign born white residents, and percent of homes reporting radio ownership.
(DOCX)

**S5 Table. Odds ratios for birth outcomes in SF, OAK and LA from unrestricted analysis.** Abbreviations: LA-Los Angeles; OAK-Oakland; SF-San Francisco; PTB-preterm birth; LBW-low birth weight; SGA-small for gestational age; PM-perinatal mortality; OR-odds ratio; CI-confidence interval.
(DOCX)

**S6 Table. Maternal race by HOLC grade and metropolitan area.** Abbreviations: N-number; %-percentage; NH-non-Hispanic.
(DOCX)

**S7 Table. 1940s areal weighted census tract metrics by HOLC grade.** Abbreviations: N-number; %-percentage; IQR-interquartile range; SD-standard deviation.
(DOCX)

## Acknowledgments

The authors would like to thank the members of the Sustainability and Health Equity (S/HE) Lab, specifically Kathy Tran for technical and data management support and Dr. Lara Cushing for methodological critique that improved the statistical analysis of this study.

## Author Contributions

**Conceptualization:** Anthony L. Nardone, Rachel Morello-Frosch.

**Data curation:** Anthony L. Nardone, Rachel Morello-Frosch.

**Formal analysis:** Anthony L. Nardone, Joan A. Casey, Kara E. Rudolph, Rachel Morello-Frosch.

**Funding acquisition:** Rachel Morello-Frosch.

**Investigation:** Anthony L. Nardone, Joan A. Casey, Rachel Morello-Frosch.

**Methodology:** Anthony L. Nardone, Joan A. Casey, Kara E. Rudolph, Deborah Karasek, Mahasin Mujahid, Rachel Morello-Frosch.

**Project administration:** Anthony L. Nardone, Rachel Morello-Frosch.

**Resources:** Rachel Morello-Frosch.

**Software:** Anthony L. Nardone, Joan A. Casey, Kara E. Rudolph, Rachel Morello-Frosch.

**Supervision:** Joan A. Casey, Rachel Morello-Frosch.

**Validation:** Joan A. Casey, Deborah Karasek, Mahasin Mujahid, Rachel Morello-Frosch.

**Visualization:** Anthony L. Nardone, Joan A. Casey, Kara E. Rudolph, Deborah Karasek, Rachel Morello-Frosch.

**Writing – original draft:** Anthony L. Nardone.

**Writing – review & editing:** Joan A. Casey, Kara E. Rudolph, Deborah Karasek, Mahasin Mujahid, Rachel Morello-Frosch.

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
