## [Decision Letter · Decision Letter 0]

5 Jun 2020

PONE-D-20-03289

Associations between historical redlining and birth outcomes from 2006 through 2015 in California

PLOS ONE

Dear Authors,

Thank you for submitting your manuscript to PLOS ONE. After careful consideration, we feel that it has merit but does not fully meet PLOS ONE’s publication criteria as it currently stands. Therefore, we invite you to submit a revised version of the manuscript that addresses the points raised during the review process.

Please submit your revised manuscript by 04 August 2020. If you will need more time than this to complete your revisions, please reply to this message or contact the journal office at plosone@plos.org. Please include the following items when submitting your revised manuscript:

We look forward to receiving your revised manuscript.

Kind regards,

Professor Hafiz T.A. Khan, Ph.D

Academic Editor

PLOS ONE

Journal Requirements:

2. ‘In ethics statement in the manuscript and in the online submission form, please provide additional information about the patient records used in your retrospective study. Specifically, please ensure that you have discussed whether all data were fully anonymized before you accessed them and/or whether the IRB or ethics committee waived the requirement for informed consent. If patients provided informed written consent to have data from their medical records used in research, please include this information.

Reviewers' comments:

Reviewer's Responses to Questions

**Comments to the Author**

1. Is the manuscript technically sound, and do the data support the conclusions?

Reviewer #1: Yes

Reviewer #2: Yes

2. Has the statistical analysis been performed appropriately and rigorously? 

Reviewer #1: Yes

Reviewer #2: Yes

3. Have the authors made all data underlying the findings in their manuscript fully available?

Reviewer #1: No

Reviewer #2: No

4. Is the manuscript presented in an intelligible fashion and written in standard English?

Reviewer #1: Yes

Reviewer #2: Yes

5. Review Comments to the Author

Reviewer #1: The article is well written. The aim is clear and the statistical treatment and interpretations are appropriate, although I am inclined to think propensity score weighting (see Austin and Stuart, 2015) would have yielded better results to matching since it does not discard any data. This does not take anything away from the statistical analysis, which is well executed and in line with the research aims but maybe something the authors might want to explore in future studies.

Reviewer #2: THis is a very interesting paper covering a very important topic. The analysis seems appropriate and the piece is well written.

However, the paper seems rather unbalanced. The introduction seems to be very short and doesn't really set up much of a theoretical framework. This means that the research questions are generally only tied into a empirical strategy. I think this is a real shame because this can link into a much broader theoretical system. This is then somewhat reflected in the (rather longer) discussion. Here, most of the focus seems to be on the USA. By using a stronger theoretical framework the implications for policy and practice internationally can be much broader.

A few style points - the acronyms become very distracting. This line from the abstract is almost impossible to read: The prevalence of PTB, SGA and PM tended to be higher in lower HOLC grades, while

the prevalence of LBW varied across grades. Overall odds of PM and PTB increased

as HOLC grade worsened.

Also many of the references appear outside of the period, rather than within it.

6. PLOS authors have the option to publish the peer review history of their article (what does this mean?). If published, this will include your full peer review and any attached files.

Reviewer #1: No

Reviewer #2: No

---

## [Author Response · Author response to Decision Letter 0]

20 Jul 2020

July 20, 2020

Professor Hafiz T.A. Khan, Ph.D Academic Editor

PLOS ONE

Dear Dr. Khan:

Thank you for the opportunity to revise our manuscript (PONE-D-20-03289), entitled, “Associations between historical redlining and birth outcomes from 2006 through 2015 in California.”

Below we provide our responses to reviewer and editor feedback.

Editor’s Comments:

Journal Requirements:

1. Please ensure that your manuscript meets PLOS ONE's style requirements, including those for file naming. The PLOS ONE style templates can be found at https://journals.plos.org/plosone/s/file?id=wjVg/PLOSOne_formatting_sample_main_body.pdf and https://journals.plos.org/plosone/s/file?id=ba62/PLOSOne_formatting_sample_title_authors_affiliations. pdf

Response: We have revised the paper to comply with PLOS ONE’s style requirements.

2. ‘In ethics statement in the manuscript and in the online submission form, please provide additional information about the patient records used in your retrospective study. Specifically, please ensure that you have discussed whether all data were fully anonymized before you accessed them and/or whether the IRB or ethics committee waived the requirement for informed consent. If patients provided informed written consent to have data from their medical records used in research, please include this information.

Response: We have revised our ethics statement in the online form to say the following: “Birth certificate data were acquired from the California Department of Public Health and study protocols, which waived requirements for informed consent, were approved by the Institutional Review Boards of the CA Department of Public Health (#13-05-z) and the University of California, Berkeley (# 2013-10- 5,693).”

3. We note that you have indicated that data from this study are available upon request. PLOS only allows data to be available upon request if there are legal or ethical restrictions on sharing data publicly. For information on unacceptable data access restrictions, please

see http://journals.plos.org/plosone/s/data-availability#loc-unacceptable-data-access-restrictions.

b) If there are no restrictions, please upload the minimal anonymized data set necessary to replicate your study findings as either Supporting Information files or to a stable, public repository and provide us with the relevant URLs, DOIs, or accession numbers. Please

see http://www.bmj.com/content/340/bmj.c181.long for guidelines on how to de-identify and prepare clinical data for publication. For a list of acceptable repositories, please

see http://journals.plos.org/plosone/s/data-availability#loc-recommended-repositories.

Response: The sharing of anonymized data from this study are restricted from transmission to those outside the study team, due to ethical and legal guidelines from the Institutional Review Boards that approved our study. However, investigators seeking access to birth certificate data can request it directly from the California Office of Statewide Health Planning and Development (OSHPD) at DataandReports@OSHPD.ca.gov. Additional information on how to request the birth certificate data can be found here: https://oshpd.ca.gov/data-and-reports/research-data-request-information/

Response: Captions have been included for Supporting Information files at the end of the manuscript.

Reviewers' comments:

Reviewer's Responses to Questions

The PLOS Data policy requires authors to make all data underlying the findings described in their manuscript fully available without restriction, with rare exception (please refer to the Data Availability Statement in the manuscript PDF file). The data should be provided as part of the manuscript or its supporting information, or deposited to a public repository. For example, in addition to summary statistics, the data points behind means, medians and variance measures should be available. If there are restrictions on publicly sharing data—e.g. participant privacy or use of data from a third party— those must be specified.

Reviewer #1: No 

Reviewer #2: No

Response: Please see our explanation above about the California Department of Public Health IRB guidelines that preclude us from sharing our birth certificate data on a public repository. We have provided contact information for researchers who wish to acquire the birth certificate data used in this study directly from the California Office of Statewide Health Planning and Development (OSHPD).

5. Review Comments to the Author

Reviewer #1: The article is well written. The aim is clear and the statistical treatment and interpretations are appropriate, although I am inclined to think propensity score weighting (see Austin and Stuart, 2015) would have yielded better results to matching since it does not discard any data. This does not take anything away from the statistical analysis, which is well executed and in line with the research aims but maybe something the authors might want to explore in future studies.

Response: In response to Reviewer #1’s comments, we have made the following improvements:

• Further elaborated on description of propensity score exclusion, matching and weighting approach taken (lines 340-349), as weighting was employed in analyses following restriction of data.

Reviewer #2: This is a very interesting paper covering a very important topic. The analysis seems appropriate and the piece is well written.

However, the paper seems rather unbalanced. The introduction seems to be very short and doesn't really set up much of a theoretical framework. This means that the research questions are generally only tied into an empirical strategy. I think this is a real shame because this can link into a much broader theoretical system. This is then somewhat reflected in the (rather longer) discussion. Here, most of the focus seems to be on the USA. By using a stronger theoretical framework the implications for policy and practice internationally can be much broader.

A few style points - the acronyms become very distracting. This line from the abstract is almost impossible to read: The prevalence of PTB, SGA and PM tended to be higher in lower HOLC grades, while the prevalence of LBW varied across grades. Overall odds of PM and PTB increased as HOLC grade worsened.

Also many of the references appear outside of the period, rather than within it.

Response: In response to Reviewer #2’s comments, we have made the following improvements:

• We added text to the introduction to expand on our theoretical framework for understanding how historical redlining can influence current health outcomes (lines 176-213). We did not expand the geographical nature of this discussion, as this redlining policy was unique to the United States.

• We added text to the discussion to incorporate more recent literature on this topic

• We removed the abbreviations for preterm birth and perinatal mortality throughout the paper

• We placed the period after our citations.

If you have further questions regarding our resubmission, please do not hesitate to contact us. 

Regards,

Rachel Morello-Frosch, Ph.D., M.P.H.

Professor, Department of Environmental Science, Policy and Management &

School of Public Health

Anthony Nardone, M.S., M.S., M.D.(c)

Medical Student, University of California Berkeley—University of California San Francisco Joint Medical Program

---

## [Editor Report · Decision Letter 1]

23 Jul 2020

Associations between historical redlining and birth outcomes from 2006 through 2015 in California

PONE-D-20-03289R1

Dear Authors,

We’re pleased to inform you that your manuscript has been judged scientifically suitable for publication and will be formally accepted for publication once it meets all outstanding technical requirements.

Kind regards,

Professor Hafiz T.A. Khan, Ph.D

Professor of Public Heath & Statistics, University of West London, UK  

Academic Editor

PLOS ONE